# Factors Influencing Morbidity and Mortality Rates in Tertiary Intensive Care Units in Turkey: A Retrospective Cross-Sectional Study

**DOI:** 10.3390/healthcare12060689

**Published:** 2024-03-19

**Authors:** Ümit Murat Parpucu, Onur Küçük, Semih Aydemir

**Affiliations:** 1Department of Anesthesiology and Reanimation, Gülhane Faculty of Health Sciences, University of Health Sciences, 06010 Ankara, Türkiye; drmuratparpucu@gmail.com; 2Department of Anesthesiology and Reanimation, Ankara Atatürk Sanatoryum Training and Research Hospital, University of Health Sciences, 06290 Ankara, Türkiye; dr.okucuk@gmail.com; 3Department of Anesthesiology and Reanimation, Yenimahalle Training and Research Hospital, University of Yıldırım Beyazıt, 06370 Ankara, Türkiye

**Keywords:** critical care, mortality, pressure injury, APACHE, intensive care units/standards, intensive care units/statistics and numerical data, nurse–patient ratios

## Abstract

Background and Objectives: The objective of this study was to determine the correlation between the prognosis of patients admitted to a tertiary intensive care unit (ICU) and the admitted patient population, intensive care conditions, and the workload of intensive care staff. Materials and Methods: This was a retrospective cross-sectional study that analyzed data from all tertiary ICUs (a minimum of 40 and a maximum of 59 units per month) of eight training and research hospitals between January 2022 and May 2023. We compared monthly data across hospitals and analyzed factors associated with patient prognosis, including mortality and pressure injuries (PIs). Results: This study analyzed data from 54,312 patients, of whom 51% were male and 58.8% were aged 65 or older. The median age was 69 years. The average number of tertiary ICU beds per unit was 15 ± 6 beds, and the average occupancy rate was 83.57 ± 19.28%. On average, 7 ± 9 pressure injuries (PI) and 10 ± 7 patient deaths per unit per month were reported. The mortality rate (18.66%) determined per unit was similar to the expected rate (15–25%) according to the Acute Physiology and Chronic Health Evaluation (APACHE) II score. There was a statistically significant difference among hospitals on a monthly basis across various aspects, including bed occupancy rate, length of stay (LOS), number of patients per ICU bed, number of patients per nurse in a shift, rate of patients developing PI, hospitalization rate from the emergency department, hospitalization rate from wards, hospitalization rate from the external center, referral rate, and mortality rate (*p* < 0.05). Conclusions: Although generally reliable in predicting prognosis in tertiary ICUs, the APACHE II scoring system may have limitations when analyzed on a unit-specific basis. ICU-related conditions have an impact on patient prognosis. ICU occupancy rate, work intensity, patient population, and number of working nurses are important factors associated with ICU mortality. In particular, data on the patient population admitted to the unit (emergency patients and patients with a history of malignancy) were most strongly associated with unit mortality.

## 1. Introduction

Tertiary intensive care units (ICUs) are highly specialized hospital environments equipped with many complex technologies [1]. They are locations where life-threatening diseases are treated and organ support is provided for invasive monitoring, thus preventing multiple-organ failure and reducing mortality [2]. The use of scoring systems to predict the risk of death and evaluate outcomes in critically ill patients is vital in modern medicine [3]. In ICUs, numerous scoring systems have been developed over the last two decades for general ICU patients or defined subgroups. Acute Physiology and Chronic Health Evaluation (APACHE) II is the scoring system developed for the general ICU population, predicting the risk of in-hospital mortality, and it is the most widely used scoring system to determine the severity of disease in the ICU [1,4]. The use of the APACHE II scoring system for ICU patients is not limited to mortality data; it has also been shown to be predictive of the development of pressure injuries (PIs) in critically ill patients [5].

Risk scoring is a highly complex system for comparing outcomes in ICUs [1]. Multiple variables are required to calculate scores for these scoring systems, and although there is difficulty in collecting data, many studies have reported the helpful performance of the APACHE II scoring system [6,7,8,9]. Although patient-based values are taken into account in all of these scoring systems, a scoring system that takes into account the conditions of hospitals, ICUs, and employees has not been prepared, and there is no study on this in the literature.

Studies have reported differences in patient outcomes between high- and low-volume hospitals, using both hospital and individual surgeon volume as the unit of analysis [10]. Hospital volume reflects institutional characteristics such as infrastructure, number of beds, occupancy rate, bed–patient ratio, and nurse–patient ratio. Surgeon volume can be considered an indicator of the surgeon’s technical or decision-making skills, which can affect patient outcomes [10,11]. The available evidence supports higher-volume hospitals for better outcomes, and this has been applied in quality and cost improvement policies over the years [10]. A literature review of 40 studies on the volume–outcome relationship in critically ill patients has shown that those admitted to high-volume hospitals have better outcomes [12]. This is particularly relevant given the current shortage of intensive care physicians and the general complexity of critical illnesses [12]. In our country, training and research hospitals (TRHs) are considered high-volume hospitals. Recently, newly established city hospitals (CHs) have been added to the existing hospitals. This policy is still supported in our country in terms of quality and cost.

The literature generally limits the relationship between intensive care working conditions and mortality and morbidity to studies examining nurse staffing levels and adverse patient outcomes [13]. Adequate nurse and staffing levels are indistinguishably linked to favorable patient outcomes both in general ward settings and in critical care areas, including the ICU [14]. Inadequate staffing levels, coupled with increasing demand for intensive care beds and decreasing budgets, can compromise patient safety [13]. Over the last decade, several studies have investigated the correlation between nurse staffing levels and patient outcomes, including mortality, complications, infection rates, PI development, falls, length of stay (LOS), and medication errors [13,15]. These studies have either focused on unit-level outcomes or aggregated their results to the hospital level, thus failing to provide a clear insight into the relationship between staffing levels and patient outcomes in the intensive care setting [13,15]. While some professional organizations have mandated a nurse–patient ratio of 1:1 [13], there is no clear international consensus on this issue. Additionally, data on tertiary ICUs, where patient care is particularly challenging, are limited in the literature.

The objective of this study was to investigate the potential relationship between patient population, intensive care conditions, and the workload of intensive care staff with mortality and PI in patients admitted to tertiary ICUs. Additionally, this study aimed to assess the suitability of the APACHE score, which is commonly used for mortality prediction in general ICUs, for use in tertiary ICUs, where critically ill patients receive the highest level of care. To enhance the generalizability of our study, we analyzed the databases of eight TRHs in Ankara, the capital of the Republic of Turkey. All of these hospitals use the same patient tracking software system, have high patient volumes, provide healthcare services in all branches, and cover 74% of the tertiary ICU beds in the region.

## 2. Materials and Methods

### 2.1. Ethical Conduct

This retrospective observational descriptive study was initiated after receiving approval from the Yıldırım Beyazıt University Yenimahalle TRH Clinical Research Ethics Committee (date: 16 August 2023; approval number: E-2023-33). All procedures followed were in accordance with the ethical standards of the committee responsible for human experimentation (institutional and national) and the Declaration of Helsinki as revised in 2013.

### 2.2. Participants

The data of all patients hospitalized in the tertiary ICUs of 8 TRHs with equal service conditions between January 2022 and May 2023 in Ankara were investigated. The only exclusion criterion was that ICU data with missing monthly data in the database were excluded from this study. CHs were also included in this study because they had TRH status. All data were collected on a monthly basis, and data from all tertiary ICUs were probed for 17 months. The data were collected in three daily shifts and entered into the system at the end of each month. Two of the hospitals were CHs, and six were TRHs. Ankara CH: 9 months (January–September 2022) 18 units, 8 months (October 2022–May 2023) 19 units; Etlik CH (ICU data were analyzed as of January 2023 because it was newly opened): 1 month (January 2023) 14 units, 4 months (February–May 2023) 19 units; Ankara TRH: 4 months (January–April 2022) 8 units, 2 months (May–June 2022) 7 units, 11 months (July 2022–May 2023) 6 units; Dışkapı Yıldırım Beyazıt TRH: 8 months (January–August 2022) 8 units, 1 month (September 2022) 7 units, 1 month (October 2022) 3 units, and 7 months (November 2022–May 2023) 1 unit; Gülhane TRH and Atatürk TRH: 17 months 5 units; Yenimahalle TRH: 8 months (January–August 2022) 2 units, 9 months (September 2022–May 2023) 3 units; Dr. Abdurrahman Yurtaslan Oncology TRH: data from 1 tertiary care ICU were examined for 17 months. Complete data were sent from ICUs and recorded in the database.

### 2.3. Interventions and Clinical Definitions

Hospitals were coded according to the order of their initials. Ankara Atatürk Sanatoryum TRH—H-1; Ankara TRH—H-2; Ankara CH—H-3; Dışkapı Yıldırım Beyazıt TRH—H-4; Dr. Abdurrahman Yurtaslan Oncology TRH—H-5; Etlik CH—H-6; Gülhane TRH—H-7; and Yenimahalle TRH—H-8.

To compare the demographic characteristics of the patients in detail, we divided ages into 5 groups: 18–44, 45–64, 64–74, 75–89, and over 89.

APACHE II is a model in which 12 physiological variables of the patient are included [4]. It gives a single score up to a maximum of 71. It is administered within 24 h of ICU admission, and the lowest value for each component of the physiology variable is recorded. Applying logistic regression calculates the individual hospital risk of death by converting the score into the probability of death. A higher score in this model indicates greater disease severity due to its impact on mortality. The APACHE II score and in-hospital mortality rate were defined in a study conducted by Knaus et al. in 1985. The in-hospital mortality rate relationship according to APACHE II score distribution is as follows: a score of “0–4” is defined as a 4% mortality rate, a score of “5–9” is defined as an 8% mortality rate, a score of “10–14” is defined as a 15% mortality rate, a score of “15–19” is defined as a 25% mortality rate, a score of “20–24” is defined as a 40% mortality rate, a score of “25–29” is defined as a 55% mortality rate, a score of “30–34” is defined as a 73% mortality rate, and a score of “over 34” is defined as an 85% mortality rate [4]. APACHE scores were divided into 8 groups: 0–4, 5–9, 10–14, 15–19, 20–24, 25–29, 30–34, and over 34.

Age, gender, and APACHE II score distributions of patients hospitalized in the tertiary care ICU were taken directly from the system. Moreover, we obtained the following data from the database: the daily number of ICU beds, number of inpatients, bed occupancy rate (monthly ratio calculated by the number of daily inpatients and the number of ICU beds), patients’ LOS (monthly average of the LOS of all inpatients), number of patients per bed (ratio of the number of monthly inpatients to the number of ICU beds), number of patients per nurse (monthly rate calculated by dividing the number of nurses working in one shift per day by the maximum number of intensive care patients admitted in one shift per day), rate of patients with PI (monthly rate calculated by the number of patients with PI and the total number of inpatients in the ICU), hospitalization rate from the emergency department (ratio of monthly number of patients admitted to ICU from the emergency department to total number of inpatients), hospitalization rate from wards (ratio of the monthly number of patients admitted to ICU from hospital wards and the total number of inpatients), hospitalization rate from external center (ratio of the monthly number of patients admitted to ICU in external centers and total number of inpatients), rate of referred patients (ratio of the monthly number of patients referred to external centers and total number of inpatients), and mortality rate (monthly number of patients with exitus in ICU and the ratio of the total number of inpatients). The data from each hospital were compared.

### 2.4. Outcomes

The primary outcome was the relationship between the mortality rate in tertiary care ICUs and the admitted patient population, ICU conditions, and the workload of ICU staff. The second outcome was the relationship between the mortality rate observed in tertiary care ICUs and the mortality rate estimated by the APACHE II score. The third outcome was the relationship between PIs detected in tertiary ICUs and the patient population, ICU conditions, and the workload of ICU staff.

### 2.5. Statistical Analysis

All data obtained and recorded during this study were analyzed using the Jamovi statistical program, version 2.3.21.0 (Sydney, Australia), and we created graphical representations. We used the Shapiro–Wilk test to assess whether the data were normally distributed. For non-normally distributed or ordinal data, we used median quartiles. Categorical variables are presented in terms of the number and percentage of cases, and we evaluated them using chi-squared and Fisher’s exact tests. Since this study included data from 8 hospitals, we analyzed continuous variables that did not comply with normal distribution using Welch’s one-way ANOVA test or the Kruskal–Wallis test. Differences between hospitals were analyzed with the Games–Howell post hoc test or the Dwass–Steel–Critchlow–Fligner pairwise comparisons test. We used Spearman’s correlation analysis to analyze the relationship between mortality rate and ICU data, as our data did not follow a normal distribution. When appropriate, we calculated 95% confidence intervals (CIs), and we considered *p*-values of less than 0.05 to be statistically significant.

Significance values were adjusted with the Bonferroni correction for multiple comparisons, and when comparing 8 hospitals, we considered *p*-values below 0.006 to be statistically significant.

## 3. Results

Data from 54,312 patients hospitalized in the tertiary care ICU in Ankara between January 2022 and May 2023 were included in this study. In the entire patient group, 51% were male and the median age was 69 years. The APACHE II score of 29% of the patients was below 10. The median APACHE II score across hospitals was 15–19, and expected mortality was 15–25%. There was no statistically significant difference in comparing the demographic characteristics of the patients according to hospitals, as demonstrated in Table 1.

The average number of tertiary care ICU beds per unit was 15 ± 6 beds, and the average occupancy rate was 83.57 ± 19.28%. The lowest occupancy rate was seen in September 2022, and there was no statistically significant difference between months in terms of bed occupancy rates (*p* = 0.618, Welch’s one-way ANOVA). The average number of tertiary ICUs from which monthly data were collected was 48, with a minimum of 40 and a maximum of 59 units. The number of inpatients per unit per month was 65 ± 39 patients, and the average ICU stay was 5.76 ± 5.84 days. Moreover, the number of nurses working per unit per month was 28 ± 10 nurses (working in three shifts). An average of 7 ± 9 PIs per month were reported per unit (a PI rate average of 11.93 ± 15.29% per unit), and an average of 10 ± 7 patient deaths were reported (a mortality rate average of 18.66 ± 15.36% per unit). The mortality rate determined per unit (an average of 18.66%) and the expected mortality rate according to the APACHE II score (15–25%) were similar.

Among hospitals on a monthly basis, there was a statistically significant difference in terms of the percentage of patients with bed occupancy, average patient LOS, number of patients per intensive care bed, number of patients per nurse in a shift, percentage of patients developing PI, emergency department patient hospitalization percentage, ward patient hospitalization percentage, outpatient center patient hospitalization percentage, referral percentage of patients, and percentage of patients with mortality (*p* < 0.001, *p* < 0.001, *p* < 0.001, *p* < 0.001, *p* < 0.001, *p* < 0.001, *p* < 0.001, *p* < 0.001, *p* < 0.001, and *p* < 0.001, respectively, according to Welch’s one-way ANOVA; Table 2). In terms of the bed occupancy percentage difference, H-5 had the highest occupancy rate, with a median of 95%. The statistical difference in terms of the average LOS originates from H-2 and H-7 hospitals, and the median LOS in these hospitals was the highest at 8. With regard to the number of patients per bed, H-5 showed a statistically significant difference, and the number of patients per bed was the lowest in this hospital. H-5 was the hospital with the highest bed occupancy rate and the lowest number of patients per bed. When the number of patients per nurse in a shift was examined across hospitals, the average was 1.3 ± 0.4. However, H-3, H-6, and H-7 exhibited statistically significant differences compared to the other hospitals, with the number of patients per nurse per shift in these hospitals being higher than in other hospitals. Considering the percentage of patients with PI, H-3 and H-7 exhibited differences, with their rates being higher than in other hospitals. H-7 was the hospital with the most extended stay, the highest number of patients per nurse, and the highest number of patients with PI. Examining the hospitalization service percentages of patients admitted to the tertiary ICU in hospitals, H-4 showed a statistically significant difference, exhibiting the highest number of patients hospitalized from the emergency department, with H-1 hospitalizing the fewest number of patients from the emergency department. While H-4 and H-8 had the lowest number of hospitalizations, H-1 had the highest number of hospitalizations in terms of referral from an external center. Observing the percentage of patients referred from an external center per unit, H-3, H-4, and H-6 exhibited statistically significant differences, with patient referral being the lowest in these hospitals. In addition, the primary reason for referral to another hospital from tertiary ICUs was the need for palliative care.

Examining mortality rates among patients (Figure 1), the mortality rate was significantly higher at H-5 compared to the other hospitals. There was a statistically significant difference between H-1 and H-2 (*p* = 0.036; Games–Howell post hoc test), H-6 (*p* = 0.009; Games–Howell post hoc test), and H-8 (*p* = 0.017; Games–Howell post hoc test), with H-1 also being high. There was a statistically significant difference between H-7 and H-6 (*p* = 0.039; Games–Howell post hoc test) and H-8 (*p* = 0.044; Games–Howell post hoc test), with the mortality rate at H-7 being high. H-5, a specific oncology hospital, followed patients with a history of malignancy. The mortality rate and patients’ history of malignancy were highly correlated.

The relationship between the mortality rate and ICU data is shown in Table 3. There was a positive correlation between the number of tertiary ICU beds, bed occupancy percentage, patient LOS, emergency room patient hospitalization rate, external center patient hospitalization rate, extended patient hospitalization rate, PI patient rate, number of patients per nurse, and mortality rate (*p*-values of <0.001, <0.001, <0.001, <0.001, 0.004, <0.001, <0.001, and <0.001, respectively). There was a negative correlation between hospitalization and mortality rate (*p* < 0.001).

The relationship between the proportion of patients with PIs and ICU data is shown in Table 4. There was a positive correlation between the number of tertiary ICU beds, bed occupancy rate, patient LOS, emergency room patient hospitalization rate, extended patient stay rate, number of patients per nurse, and the rate of patients with PI (*p*-values of <0.001, <0.001, <0.001, 0.004, <0.001, <0.001, and <0.001, respectively). There was a negative correlation between the ward patient hospitalization rate, the external center patient hospitalization rate, and the PI incidence (*p* values of <0.001 and <0.001, respectively).

## 4. Discussion

In the results of our study, we examined the data of 54,312 patients. We analyzed the data from all tertiary ICUs in eight training and research hospitals with similar facilities, serving as last-resort centers for critically ill patients. We found that the median (15–25%) mortality data estimated according to the APACHE II score and the average mortality per unit (18.66%) were similar. We found that the gender, age, and APACHE scores of patients hospitalized in the ICUs were similar. However, there was no statistical difference in the demographic characteristics of patients hospitalized in ICUs between hospitals, ICU occupancy rate, patient LOS, number of patients per bed, number of patients per nurse, rate of patients with PI, emergency department patient admission rate, and ward patient admission rate; we found that external center patient hospitalization, referral patient, and patient mortality rates were statistically significantly different between hospitals. When we examined the data related to mortality, we detected that mortality increases with the increase in the number of tertiary ICU beds, bed occupancy percentage, patient hospitalization length, emergency room patient hospitalization rate, external center patient hospitalization rate, extended patient hospitalization rate, PI patient rate, and the number of patients per nurse. We discovered that the mortality rate decreased as the hospitalization rate increased. Moreover, we also examined the ICU conditions associated with the incidence of PIs, causing an increased risk of infection in patients, which is the most important factor of mortality in critically ill patients. As the number of tertiary ICU beds, bed occupancy rate, patient LOS, emergency room patient hospitalization rate, extended patient stay rate and the number of patients per nurse increased, the rate of PI patients in the ICU also increased. The PI patient rate decreased when the ward patient hospitalization and external center patient hospitalization rates increased.

Several expert working groups supported by the National Institutes of Health and the Societies of Critical Care Medicine have recommended the regionalization of critical care medicine [7,8]. There is ample evidence showing that hospitals and physicians with high patient volumes experience better patient outcomes across a wide range of medical conditions and surgical procedures [8,16]. However, as there is not yet a study in the literature comparing these high-volume hospitals, there is no study on whether ICU conditions are related to mortality, despite the increased use of technology and improved healthcare conditions. Tertiary ICUs have many standard features, but the organization and delivery of intensive care services vary [8,17]. In our study, we examined tertiary ICU data from eight training and research hospitals. Although patient characteristics and facilities were similar, there was a statistically significant difference in patient mortality between hospitals. The mortality rate in the Dr. Abdurrahman Yurtaslan Oncology TRH ICU (a median of 57.7%) was significantly higher than in other hospitals. We attributed this high rate to the fact that this hospital’s patient demographics differ from that of the other hospitals. Since there is no separate scoring for malignancy in the APACHE II score, although the prediction score between hospitals was similar, the mortality rate was found to be significantly high in oncology hospitals due to the large patient population with a history of malignancy. There is a separate score for malignancy in the newly defined APACHE IV score in the literature, and the use of this scoring system is limited today. We concluded that the APACHE IV scoring system will, thus, provide more accurate results in predicting ICU mortality.

Prognosis in critically ill patients is related to many risk factors, such as age, gender, disease severity, comorbidities, diagnosis, and response to treatment [18,19]. Unit-derived clinical outcomes have increased the need for outcome review and guidance on the effective use of services [20,21]. Scoring systems can be used to estimate expected mortality, adjusted for differences in diagnoses, physiological abnormalities, and outcomes of critically ill patients admitted to the ICU [6,22]. Therefore, global disease severity scoring systems have grown in popularity, allowing an international comparison of intensive care outcomes. Although there are adversities in using risk adjustment methods to compare outcomes across ICUs, many studies have reported that APACHE II is the most appropriate scoring system for critically ill patients [6,23]. In our study, we used the APACHE II scoring system for interhospital mortality classification. There was no statistical difference in APACHE II values between hospitals. When hospital and unit-related results were evaluated, a statistical difference in mortality rates was detected between hospitals. This result clearly shows that the conditions in the ICU have an effect on mortality that is independent of patient-related values. Lapichino et al., in their study, conducted in ICUs regardless of level, found a direct relationship between mortality and intensive care occupancy rate [20]. In our study, we found a direct relationship between mortality and ICU occupancy. Flabouris et al. found that patients admitted to the ICU from emergency departments and external centers had high hospital mortality rates and extended intensive care stays [24]. In our study, increased mortality was observed in patients admitted from the emergency department and from an external center. The highest association with mortality was found in patients admitted from the emergency department. It was also found that mortality was significantly reduced in patients admitted from the ward. Although there is limited data in the literature examining the relationship between site of admission and mortality, there is no scoring system that takes these data into account. As we found in our multicenter study with a large group of patients, we believe that these data should be taken into account and that prospective studies are needed in this regard.

Nursing staffing levels in the ICU are different from those on wards and other hospital services for many reasons. The nurse–patient ratio is important in the ICU because of the need for nursing care, continuous monitoring and supervision of patients [13]. For tasks that require more than one nurse, or in situations of sickness, it may be necessary to use floating or on-call nurses to support the nurse [14]. In addition, the total number of ‘staff per bed’ working in ICUs is higher, because the same number of nursing staff must be available 24 h a day in ICUs, as opposed to the often-reduced staffing levels in other departments during night shifts. Although there is no clear consensus in the literature, a 1:1 nurse–patient ratio is recommended in ICUs [13]. There are also conflicting results in the literature about the mortality rates associated with the number of patients per nurse. Three studies [25,26,27] showed a statistically significant association between increasing nurse staffing levels and decreasing mortality rates, while four studies [28,29,30,31] found no statistically significant association. In our study, which was conducted with a monthly average of 48 tertiary ICUs, the average number of patients per nurse across hospitals was 1.3, which was higher than the recommended value in the literature. It was observed that mortality increased linearly in hospitals as the number of patients per nurse increased. We believe that the number of nurses in tertiary ICUs is important in terms of patient prognosis and that further studies are needed for international standardization.

Globally, mortality rates of patients admitted to ICUs have decreased over the last two decades [17]. This is remarkable considering the age of critically ill patients upon hospital admission, the number of comorbidities, and disease severity [32]. The mortality rate per ICU determined in our study was 18.6%, and this rate is similar to previously published studies [33]. Additionally, the average ICU stay per unit was 5.76 ± 5.84 days, being consistent with previous studies [34]. Although the average data in our study were found to be compatible with the literature, there were statistically significant differences when evaluated among the hospitals. At the same time, a significant number of intensive care patients are transferred between clinics and hospitals. Annually, it is estimated that 11,000 (a referral patient rate of 6.5% per unit) patients are transferred between hospital ICUs in the United Kingdom [35]. A significant portion of interhospital transfer occurs simply due to insufficient resources (number of beds, nurses, and staff) rather than the need to access a specific service unavailable in the referring unit [36]. In our study, the referral rate per unit was 2.3%. The most common reason for referral was a lack of hospital resources due to the need for palliative care. Our reasoning for referral is compatible with the literature. We attributed the low referral rate in our study to the fact that the hospitals included in this study were training and research hospitals and, therefore, had the best facilities. In addition, when patient admission and referral rates were compared between hospitals, the tertiary ICUs of CHs were the most appropriate units, as they had the lowest referral rates to external centers and the lowest number of patients admitted from external centers, despite their high occupancy rates. We attributed this to the recent opening of CHs and the fact that they have a higher number of tertiary ICUs due to their larger hospital areas. Given these referral rates, we believe that the number of hospital-based ICUs will continue to increase in the future due to demand.

PI is one of the most common health conditions worldwide and comprises local tissue damage caused by the compression of underlying tissues [37]. As reported by the Agency for Healthcare Research and Quality, approximately 2.5 million people are affected by PIs yearly, and more than 60,000 patients die from direct PIs each year in the United States alone [38]. The incidence of PI is high in weak, elderly, bedridden, and malnourished patients who are in prone positions, as well as in patients who are unable to care for themselves and, in particular, critically ill [39]. Many studies have been conducted on ICU risk factors. However, the relationship between PIs and critically ill patients has not been fully elucidated [5]. There are many predictive scales for essential mortality factors in intensive care patients, such as mortality and delirium prediction scales [6,40]. There is no PI risk assessment and prediction scale explicitly used for intensive care patients [41]. In 2022, an extensive systematic review and meta-analysis study conducted by Wen Tang et al. demonstrated that the APACHE II scoring system was a significant determinant of PIs in the ICU and found a high correlation between high APACHE score and the development of PIs in the ICU [5]. Our study examined the relationship between PIs and unit-based data. We found a linear relationship between the number of tertiary ICU beds, bed occupancy rate, patient LOS, emergency room patient hospitalization rate, extended patient hospitalization rate, number of patients per nurse, and the rate of patients with PI. A statistically significant inverse relationship existed between the ward patient hospitalization rate, external center patient hospitalization rate, and PI incidence. There was a significant correlation between the number of patients per nurse and the development of PIs. In addition, ICU mortality increased as the number of patients with PIs increased. In the light of these results, we can say that staff and unit conditions in tertiary ICUs are closely related to inpatient prognosis. Studies on unit-based data and PIs are limited in the literature, and we believe extensive prospective studies are needed.

Our study had several limitations. Firstly, this study examined data from tertiary ICUs of TRHs in Ankara, which had the same conditions and used the same database. Data from both university hospitals and private hospitals were unavailable. However, as this study includes data from 74% of tertiary intensive care beds in the region, it can be inferred that the study results are representative of the general population. Secondly, data on the number of doctors and staff working in ICUs was not available. The quantity of working doctors could impact mortality rates in ICU, while the number of active staff members may play a role in the development of PIs. Thirdly, the data were obtained from tertiary ICU databases at hospitals, and its accuracy was deemed reliable. However, individual patient records were not assessed. Due to the long follow-up period of 17 months and analysis of an average of 48 tertiary ICU data per month, it can be concluded that the study results were minimally affected by potential data bias. Another limitation of this study is that nursing workload is based on the number of patients per nurse. To provide a more comprehensive evaluation, prospective observational studies using other scoring systems, such as the Nurse Activity Index, are needed. Additionally, this study did not take into account the patients’ skill mix upon admission to the ICUs, nor did it consider whether they were receiving respiratory support. These factors could potentially impact the workload of ICU and ultimately affect patient prognosis.

## 5. Conclusions

Tertiary care ICUs of training and research hospitals are units with high patient volumes, ample facilities and resources, and better patient outcomes. Although these units have many standard features, the organization and presentation of intensive care services vary. Resource use and inpatient prognosis data are not the same in these units, which represent the most advanced level.

There are many highly validated scoring systems used in the literature to predict prognosis in ICUs; however, none of them include ICU and staff conditions. Although these scoring systems are successful as general data, they have shortcomings when examined on a unit basis. ICU conditions have a significant impact on patient prognosis. ICU occupancy, work intensity, patient population and the number of nurses working in the ICU are important mortality factors. In particular, the patient population admitted to the unit (emergency patients and history of malignancy) is the data most strongly associated with unit mortality. At the same time, the development of PIs, which is associated with ICU mortality, is closely related to ICU and staffing conditions. Although more technologically advanced and larger centers are being established in today’s world, where the population is growing and the need for intensive care continues to increase, international standardization of these centers in terms of working conditions is essential. Further prospective studies examining the effect of unit-related conditions on mortality are needed.

## Figures and Tables

**Figure 1 healthcare-12-00689-f001:**
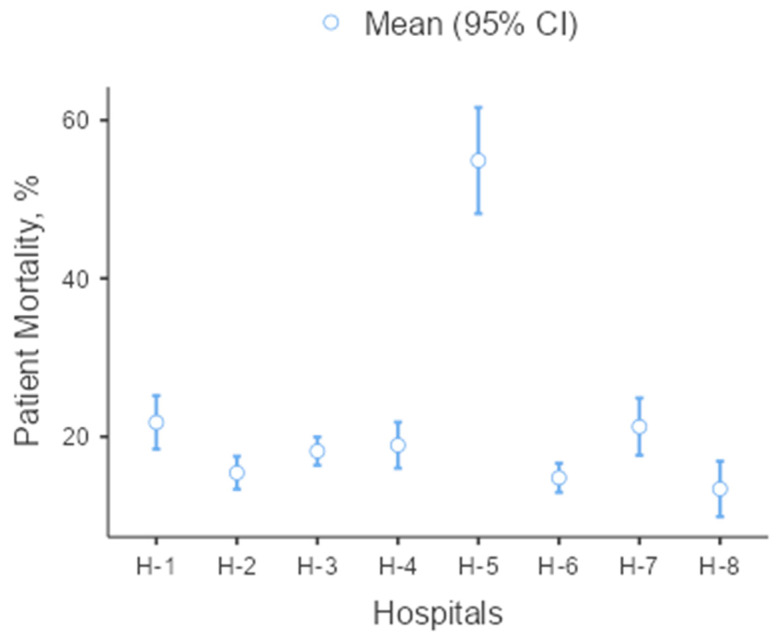
Mortality rates between hospitals.

**Table 1 healthcare-12-00689-t001:** Comparison of demographic characteristics of patients according to hospitals.

	H-1	H-2	H-3	H-4	H-5	H-6	H-7	H-8	*p*-Value
	N = 5492	N = 3799	N = 24,945	N = 4093	N = 435	N = 7711	N = 5980	N = 1857	
Gender (%)									0.977 ^†^
Female	50.5%	48.3%	50.2%	48.6%	47.5%	49.6%	46.4%	47.2%	
Male	49.5%	51.7%	49.8%	51.4%	52.5%	50.4%	53.6%	52.8%	
Age	69	70	69	68.5	60.5	70	69	71	0.080 *
Median (Q1–Q3)	(51–79)	(55–81)	(50–81)	(49–81)	(49–74)	(51–82)	(52–80)	(52–82)	
Age (%)									0.088 ^†^
18–44	15.3%	16.2%	19.5%	18.6%	20.0%	18.7%	18.4%	18.8%	
45–64	26.5%	22.3%	22.1%	22.9%	33.8%	20.9%	24.1%	20.1%	
65–74	24.0%	22.3%	22.1%	21.4%	22.5%	21.3%	23.8%	21.5%	
75–89	25.2%	26.0%	19.8%	22.9%	18.8%	20.9%	23.8%	26.4%	
Over 89	9.0%	13.2%	16.5%	14.3%	5.0%	18.2%	10.0%	13.2%	
APACHE score (%)									0.999 ^†^
0–4	9.3%	13.6%	12.1%	5.7%	7.5%	13.3%	16.1%	9.7%	
5–9	17.4%	17.7%	17.7%	17.1%	12.5%	16.0%	14.6%	20.8%	
10–14	15.9%	14.3%	15.1%	20.0%	15.0%	15.8%	14.2%	12.5%	
15–19	15.3%	14.3%	14.7%	17.1%	15.0%	14.0%	13.0%	15.3%	
20–24	13.1%	10.6%	12.6%	12.9%	13.8%	12.2%	13.0%	12.5%	
25–29	10.0%	10.2%	10.5%	11.4%	15.0%	10.7%	11.5%	11.8%	
30–34	10.0%	9.1%	8.6%	7.1%	12.5%	9.8%	8.8%	6.9%	
Over 34	9.0%	10.2%	8.8%	8.6%	8.8%	8.2%	8.8%	10.4%	

Categorical variables are expressed as either frequency (n) or percentage (%). Categorical variables were compared using Pearson’s chi-square test or Fisher’s exact test. ^†^ Continuous variables are expressed as median quartiles (Q1–Q3). Continuous variables were compared with Welch’s and Fisher’s one-way ANOVA tests or the Kruskal–Wallis test. * Statistically significant *p*-values are shown in bold. N: total number of inpatients.

**Table 2 healthcare-12-00689-t002:** Comparison of clinical data of hospitals.

	H-1	H-2	H-3	H-4	H-5	H-6	H-7	H-8	*p*-Value
Bed occupancy (%)	92.1	91.5	92.0	86.4	**95.0**	94.5	93.9	80.0	**<0.001 ***
Median (Q1–Q3)	(82.3–96.4)	(86.6–98)	(67–98)	(69–96)	**(92–97)**	(82.9–97.7)	(82.4–98)	(71.2–88.1)	
Patient length of stay (days	5	**8**	3	3	1	1	**8**	7	**<0.001 ***
Median (Q1–Q3)	(4–7.2)	**(6.1–10.9)**	(3–4)	(2.9–6)	(1–1)	(1–5.8)	**(4.7–11.7)**	(4.8–9.2)	
Number of patients per bed	5.3	3.7	3.0	4.1	**2.4**	4.9	3.5	4.5	**<0.001 ***
Median (Q1–Q3)	(3.8–7.2)	(2.9–4.7)	(2.6–4.4)	(2.6–8.2)	**(1.6–2.5)**	(3.6–6.4)	(2.8–6.7)	(3.7–6.7)	
Number of patients per nurse	1.2	1.2	**1.4**	1.1	0.7	**1.4**	**1.3**	1.0	**<0.001 ***
Median (Q1–Q3	(1–1.3)	(1.1–1.3)	**(1.1–1.7)**	(0.8–1.3)	(0.6–1)	**(1.2–1.5)**	**(1.2–1.4)**	(0.9–1.2)	
Patient with pressure injury (%)	2.7	1.02	**14.6**	2.9	3.8	4.9	**20.6**	0.7	**<0.001 ***
Median (Q1–Q3)	(0–6.4)	(0–3)	**(4.9–31.2)**	(0–4.7)	(0–6.7)	(2–8.2)	**(9–33.3)**	(0–2)	
Hospitalization from emergency department (%)	**17.4**	27.3	50.0	**70.1**	34.0	45.0	42.1	51.1	**<0.001 ***
Median (Q1–Q3)	**(9–26.9)**	(11–69.7)	(15.6–67.4)	**(50–89.3)**	(26.6–39.7)	(25.3–58.5)	(8.4–55.6)	(35–65.6)	
Hospitalization from ward (%)	56.7	67.8	49.0	**29.4**	59.0	54.1	57.9	**31.1**	**<0.001 ***
Median (Q1–Q3)	(38.5–77.5)	(28.9–85.2)	(32.3–84.4)	**(10.7–46.7)**	(51–67)	(40.7–74.7)	(44.3–90.9)	**(17.4–44.5)**	
Hospitalization from external center (%)	**20.0**	0	0	0	6.7	0	0	0	**<0.001 ***
Median (Q1–Q3)	**(9.8–31.6)**	(0–3.8)	(0–0)	(0–2.7)	(3–9.8)	(0–0)	(0–0)	(0–45.3)	
Referred patient (%)	1.8	2.3	**0**	**0**	7.6	**0**	2.6	3.3	**<0.001 ***
Median (Q1–Q3)	(0–3.2)	(0–5.4)	**(0–2.6)**	**(0–0)**	(3.5–10.9)	**(0–0)**	(0–5.5)	(0–5.8)	
Mortality patient (%)	**21.1**	14.1	16.7	15.6	**57.7**	14.4	**19.6**	10.1	**<0.001 ***
Median (Q1–Q3)	**(8–33.3)**	(7.9–19.4)	(6.6–26.7)	(8.2–27.9)	**(47.9–61.8)**	(8.5–21)	**(7–32.2)**	(5.1–19.7)	

Continuous variables are expressed as median quartiles (Q1–Q3). Continuous variables were compared with Welch’s and Fisher’s one-way ANOVA tests or the Kruskal–Wallis test. * Statistically significant *p*-values are shown in bold. Significance values were adjusted with the Bonferroni correction for multiple comparisons, and when comparing eight hospitals, we considered *p*-values below 0.006 to be statistically significant. The *p* significance value was corrected, and *p* < 0.05 was considered significant.

**Table 3 healthcare-12-00689-t003:** Relationship (correlation) between mortality rate and variables.

	R	*p*-Value
Number of ICU beds	0.156	**<0.001** **
Bed occupancy rate	0.378	**<0.001** **
Patient length of stay	0.356	**<0.001** **
Hospitalization rate from emergency department	0.411	**<0.001** **
Hospitalization rate from wards	−0.440	**<0.001** **
Hospitalization rate from external center	0.126	**0.004** **
* Extended hospitalization rate	0.386	**<0.001** **
Pressure injury patient rate	0.404	**<0.001** **
Number of patients per nurse	0.306	**<0.001** **

* Rate of patients staying in ICU for more than 15 days. ** The relationship between significant values was evaluated with the Spearman correlation test. Statistically significant *p*-values are shown in bold. ICU: intensive care unit, R: correlation coefficient.

**Table 4 healthcare-12-00689-t004:** Relationship (correlation) between the proportion of patients with pressure injuries and the variables.

	R	*p*-Value
Number of ICU beds	0.520	**<0.001** **
Bed occupancy rate	0.327	**<0.001** **
Patient length of stay	0.198	**<0.001** **
Hospitalization rate from emergency department	0.322	**<0.001** **
Hospitalization rate from ward	−0.251	**<0.001** **
Hospitalization rate from external center	−0.243	**<0.001** **
* Extended hospitalization rate	0.501	**<0.001** **
Number of patients per nurse	0.445	**<0.001** **

* Rate of patients staying in ICU for more than 15 days. The relationship between significant values were evaluated with the Spearman correlation test. ** Statistically significant *p*-values are shown in bold. ICU: intensive care unit, R: correlation coefficient.

## Data Availability

The datasets used and/or analyzed during the current study are available from the corresponding author upon reasonable request.

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
