# Peer review of "Factors Influencing Morbidity and Mortality Rates in Tertiary Intensive Care Units in Turkey: A Retrospective Cross-Sectional Study"

_healthcare, 2024, doi:10.3390/healthcare12060689_

Round 1
Reviewer 1 Report
Comments and Suggestions for Authors
Dear authors,
I read with interest your publication related to the impact of intensive care unit conditions and workload on patient prognosis. In this paper, you reported data from all tertiary ICUs of 8 training and research hospitals in Ankara province (Turkey).
I carefully read your paper and my comments are:
1) The detailed description of the APACHE II score should be moved to the methods section;
2) It would be useful to add in the introduction that patients admitted to research hospitals have better outcomes compare to other hospitals;
3) Age is a continuos data and should not be reported as a category. The most appropriate form would be to indicate the mean age and standard deviation.
4) In the limitations section, it should be noted that nursing workload can be measured not only by the number of patients per nurse but also with other scoring systems such as the NAS (Nurses Activity Index).
5) Another limitation to mention is that the study does not take into account the skill mix of patients admitted to the ICUs included in the study.
Author Response
Dear Editor,
We thank the reviewers for their valuable comments on the manuscript, and we have edited the manuscript to address their concerns. We have highlighted in red all the changed places in the main text. We would like to state that all the reviewers' comments have been taken into account and the necessary corrections have been made.
We hope that the article is now suitable for publication.
Yours sincerely
Semih AYDEMİR, Dr.
Yıldırım Beyazıt University Yenimahalle Teaching and Research Hospital, Department of Anaesthesiology and Resuscitation, Yenimahalle, Ankara, Turkey
On behalf of all authors.
Reply to Reviewer-1:
Dear authors,
I read with interest your publication related to the impact of intensive care unit conditions and workload on patient prognosis. In this paper, you reported data from all tertiary ICUs of 8 training and research hospitals in Ankara province (Turkey).
I carefully read your paper and my comments are:
1) The detailed description of the APACHE II score should be moved to the methods section;
A-1) The detailed explanation of the APACHE II score has been moved from the introduction section to the method section based on valuable suggestions.
2) It would be useful to add in the introduction that patients admitted to research hospitals have better outcomes compare to other hospitals;
A-2) In accordance with your suggestion, we have added annotated literature data to the introduction section. This data highlights the improved patient outcomes in research hospitals and high-volume hospitals. Thank you. We believe this addition is highly appropriate.
3) Age is a continuos data and should not be reported as a category. The most appropriate form would be to indicate the mean age and standard deviation.
A-3) Within the scope of your valuable suggestion, the age groups of hospital-based patients were given numerically and compared statistically. The results were shared both in Table 1 and in the results section. The reason why we categorized the ages of the patients was to see the distribution of patients according to hospital-based age groups in more detail and to show that the inpatients were similar between hospitals.
4) In the limitations section, it should be noted that nursing workload can be measured not only by the number of patients per nurse but also with other scoring systems such as the NAS (Nurses Activity Index).
A-4) Your suggestion has been added in the limitations section of our study, with an emphasis on the need for prospective studies on this subject. Thanks.
5) Another limitation to mention is that the study does not take into account the skill mix of patients admitted to the ICUs included in the study.
A-5) We added your valuable suggestion to the limitations section of our study. Thank you for your contribution to our article.

Reviewer 2 Report
Comments and Suggestions for Authors
Thankyou for the opportunity to review your manuscript.
It is well written and provides deep insight into factors effecting patient outcomes in ICU.
Some minor suggestions:
Title.
Im not sure 'conditions and workload' is the right term for what you are exploring. Something like 'Factors impacting morbidity and mortality across Intensive Care Units in Turkey' would give readers a better idea of what your manuscript is about.
Abstract.
L21 mentions statistically significant differences - an understanding of the size of this significance is useful for your primary outcome. By expanding this you can remove the the correlation sentence starting at L25 as this is not statistically significant.
Introduction.
A little disjointed from the primary outcome. There is a lot of content around Apache but this only forms part of your secondary outcomes. I would like to see more around reported M&M in ICUs internationally (or the absence of, where this research fills a gap).
L78 the Ankara Provincial Health Directorate will be unfamiliar with international readers - what geography does this directorate cover?
Materials and Methods
L80 should be 2.1 Ethical Conduct as its not study design
L93-101 is confusing. Why werent all units measured at the same time? Its unclear how you collected this data eg which months, years for each unit. Additionally, what is the justification for the timeframe that you used?
L112 Can you please confirm if bed occupancy is the same as number of beds staffed? Eg the bed occupancy might be 10 patients for 20 beds but at the time, only enough staff for 15 beds.
Results
Can you include how you reach your 'inclusion ICUs' ie total number of ICUs, number of ICUs with missing monthly data, final number of ICUs included.
As you can appreciate, not all patients in ICU require the same level of nursing care. Was there a breakdown of intubated vs non-intubated patients when comparing nursing ratios?
L218 ideally is in the discussion ie explaining the results.
Discussion
More discussion around your results is required - most of the current discussion could easily be moved to the introduction as it describes the current literature around the world. Readers will be interested in how your data compares to this, in addition to some ideas on what the data represents.
Conclusion.
Focus on the primary outcome needs to be more apparent
Comments on the Quality of English LanguageMinor grammatical errors only
Author Response
Dear Editor,
We thank the reviewers for their valuable comments on the manuscript, and we have edited the manuscript to address their concerns. We have highlighted in red all the changed places in the main text. We would like to state that all the reviewers' comments have been taken into account and the necessary corrections have been made.
We hope that the article is now suitable for publication.
Yours sincerely
Semih AYDEMİR, Dr.
Yıldırım Beyazıt University Yenimahalle Teaching and Research Hospital, Department of Anaesthesiology and Resuscitation, Yenimahalle, Ankara, Turkey
On behalf of all authors.
Reply to Reviewer-2:
Thank you for the opportunity to review your manuscript.
It is well written and provides deep insight into factors effecting patient outcomes in ICU.
Some minor suggestions:
Title.
I’m not sure 'conditions and workload' is the right term for what you are exploring. Something like 'Factors impacting morbidity and mortality across Intensive Care Units in Turkey' would give readers a better idea of what your manuscript is about.
A-1) Thank you for your suggestion. We have revised the title of our study to 'Factors influencing morbidity and mortality rates in tertiary intensive care units in Turkey: a retrospective cross-sectional study', as you recommended, to attract more attention. We mention this because our study focuses on tertiary care intensive care units.
Abstract.
L21 mentions statistically significant differences - an understanding of the size of this significance is useful for your primary outcome. By expanding this you can remove the the correlation sentence starting at L25 as this is not statistically significant.
A-2) Thank you for your suggestion. We have revised the result section in the summary as you requested. We believe it is now more comprehensible and clear.
Introduction.
A little disjointed from the primary outcome. There is a lot of content around Apache but this only forms part of your secondary outcomes. I would like to see more around reported M&M in ICUs internationally (or the absence of, where this research fills a gap).
A-3) Thank you for your valuable suggestion. We have revised the introduction of our study to align with the primary outcome. The description of Apache has been moved to the methods section of the study.
L78 the Ankara Provincial Health Directorate will be unfamiliar with international readers - what geography does this directorate cover?
A-4) Your suggestion is a very appropriate observation and has been corrected and expressed more clearly. We thank you.
Materials and Methods
L80 should be 2.1 Ethical Conduct as its not study design
A-5) The correction has been made according to your suggestion. We thank you.
L93-101 is confusing. Why werent all units measured at the same time? Its unclear how you collected this data eg which months, years for each unit. Additionally, what is the justification for the timeframe that you used?
A-6) Thank you for your evaluation. Unit data were collected and evaluated over a 17-month period simultaneously, starting in January 2022. The number of units changed due to newly opened and closed units. The material and methods section compiles unit data on a monthly basis, beginning in January. These changes have been detailed as per your suggestion. In the Materials and Methods section, we included an explanation that the data was collected in three shifts per day. Since Etlik City Hospital was newly opened as a hospital, data collection started later than other hospitals. We would like to say that we also mentioned this. We also explicitly stated that we did not evaluate that unit for the month due to missing data, even for a single day, and we included this in the exclusion criteria of our study. To prevent confusion regarding time, we have included the months mentioned as an explanation, as per your suggestion. Thank you.
L112 Can you please confirm if bed occupancy is the same as number of beds staffed? Eg the bed occupancy might be 10 patients for 20 beds but at the time, only enough staff for 15 beds.
A-7) We can definitely say. We stated in the methods section that our data was collected on a daily shift basis. We would like to point out that we have written more clearly and in detail in order to correct the misunderstanding in your suggestion. In addition, how all our monthly data are calculated is stated in the methods section. We would like to state that we have revised it more clearly in line with your suggestion.
Results
Can you include how you reach your 'inclusion ICUs' ie total number of ICUs, number of ICUs with missing monthly data, final number of ICUs included.
A-8) Thank you for your suggestion. Since we have a large number of intensive care units, we have stated the number of intensive care units included in the hospitals month by month, upon your request, in the methods section of our study. In addition, in the results section, it was stated in our first unrevised article that a minimum of 40 and a maximum of 59 intensive care patient data were received per month. We also added the monthly average number of units as per your suggestion. In order to make the study more readable and understandable, we think that the number of monthly hospital-based units we have written in detail with your suggestion in the methods section will be sufficient.
As you can appreciate, not all patients in ICU require the same level of nursing care. Was there a breakdown of intubated vs non-intubated patients when comparing nursing ratios?
A-9) We would like to point out that your valuable suggestion is very important. We would like to point out that we do not have enough data on this subject. For this reason, we would like to state that we express this deficiency of our study in the limitations section.
L218 ideally is in the discussion ie explaining the results.
A-10) The suggestion provided is appropriate and correct. First of all, we thank you for that. The specified section has been relocated to the discussion section.
Discussion
More discussion around your results is required - most of the current discussion could easily be moved to the introduction as it describes the current literature around the world. Readers will be interested in how your data compares to this, in addition to some ideas on what the data represents.
A-11) Thank you very much for your valuable suggestion. We expanded the discussion part of our study in line with your suggestion and made further additions to the results we found and the interpretation of these results. We think that thanks to these additions, our study will be more understandable and will contribute more to the literature and shed light on future studies. Thank you very much for your valuable contribution.
Conclusion.
Focus on the primary outcome needs to be more apparent
A-12) The conclusion section of our study has been expanded to support the primary outcome, based on your suggestion.

Reviewer 3 Report
Comments and Suggestions for Authors
Dear authors,
This manuscript presents a comprehensive and detailed retrospective cross-sectional study investigating the impact of intensive care unit (ICU) conditions and workload on patient prognosis. While the research appears methodologically sound and offers valuable insights, there are several areas where improvements could enhance clarity, depth, and relevance. Below are suggestions for improvement:
1. Abstract and Introduction Clarity:
- Abstract: Provide a more concise summary of the essential findings and their implications. The current abstract is detailed but could be streamlined to emphasize the main results and their significance clearly.
- Introduction: Increase the focus on the gap in the literature that this study aims to fill. Highlighting the unique contribution of this research early on will help to contextualize its importance.
2. Methodological Details:
- Population and Setting Description: While the study design is described, including more details about selecting the eight training and research hospitals could add value. Discussing why these particular hospitals were chosen and whether they represent tertiary ICUs, in general, would provide context.
- Statistical Analysis: The statistical methods are well-detailed, but clarifying why certain tests were chosen over others for specific data types could help readers understand the analytical decisions better.
3. Results Presentation:
- Discussion of Anomalies or Outliers: If there were any unexpected results or outliers in the data, discussing these specifically could provide deeper insights into the ICU conditions and their impacts on patient prognosis.
4. Discussion and Conclusion Enhancements:
- Broader Implications: Expand the discussion on how these findings can influence ICU practices, policy-making, and future research. Providing specific recommendations based on the results could make the conclusions more impactful.
- Limitations: While some limitations are mentioned, a more thorough examination of potential biases, the generalizability of the findings, and how these limitations could affect the interpretation of the results would strengthen the manuscript.
- Future Research Directions: Suggest specific areas for future research that build directly on the findings of this study. This could include prospective studies, research in different geographic settings, or investigations into interventions to mitigate the adverse effects of high ICU workload on patient outcomes.
5. Technical and Language Corrections:
- Consistency in Terminology: Ensure consistency in using terms and definitions throughout the manuscript. For example, clarify if "tertiary ICUs," "training and research hospitals," and "City hospitals (CHs)" are used interchangeably or signify different categories.
- Grammar and Style: A thorough proofreading to correct grammatical errors and improve sentence structure would enhance readability. Ensuring the manuscript adheres to the journal's style guide is also crucial.
6. References and Citations:
- Updating References: Check that all references are up-to-date and relevant. Including more recent studies, where available, could strengthen the argument and show the manuscript's current relevance.
- Citation Consistency: Verify that all citations are consistent in format and that all referenced works are included in the bibliography.
Implementing these suggestions could significantly enhance the manuscript's clarity, depth, and impact, making its contributions to the field more pronounced and actionable.
Yours sincerely
Comments on the Quality of English LanguageDear authors
As I mentioned regarding technical and language adjustments:
Term Uniformity: Maintaining uniformity in the terminology and definitions used across the manuscript is essential. For instance, it should be made clear whether terms like "tertiary ICUs," "training and research hospitals," and "City hospitals (CHs)" have the same meaning or represent distinct categories.
Proofreading and Formatting: Conducting a comprehensive review to rectify grammatical mistakes and refine the sentence construction will boost the manuscript's clarity. It's equally essential to ensure compliance with the journal's formatting guidelines.
Yours sincerely
Author Response
Dear Editor,
We thank the reviewers for their valuable comments on the manuscript, and we have edited the manuscript to address their concerns. We have highlighted in red all the changed places in the main text. We would like to state that all the reviewers' comments have been taken into account and the necessary corrections have been made.
We hope that the article is now suitable for publication.
Yours sincerely
Semih AYDEMİR, Dr.
Yıldırım Beyazıt University Yenimahalle Teaching and Research Hospital, Department of Anaesthesiology and Resuscitation, Yenimahalle, Ankara, Turkey
On behalf of all authors.
Reply to Reviewer-3:
Dear authors,
This manuscript presents a comprehensive and detailed retrospective cross-sectional study investigating the impact of intensive care unit (ICU) conditions and workload on patient prognosis. While the research appears methodologically sound and offers valuable insights, there are several areas where improvements could enhance clarity, depth, and relevance. Below are suggestions for improvement:
- Abstract and Introduction Clarity:
- Abstract:Provide a more concise summary of the essential findings and their implications. The current abstract is detailed but could be streamlined to emphasize the main results and their significance clearly.
A-1) Thank you for your suggestion. The conclusion section of the abstract has been revised to emphasize the main results of the study.
- Introduction:Increase the focus on the gap in the literature that this study aims to fill. Highlighting the unique contribution of this research early on will help to contextualize its importance.
A-2) Thank you for your suggestion. We have revised the introduction of our study, removed data unrelated to the main subject, and improved its comprehensibility.
- Methodological Details:
- Population and Setting Description:While the study design is described, including more details about selecting the eight training and research hospitals could add value. Discussing why these particular hospitals were chosen and whether they represent tertiary ICUs, in general, would provide context.
A-3) Thank you for your valuable suggestion. We have emphasized the reasons why these 8 hospitals were chosen and their conditions both at the end of the introduction section and in the methods section, within the scope of your suggestion. It is worth noting that we included the intensive care unit data of 8 hospitals in the capital of the Republic of Turkey, where all branches have standardized conditions, in order to increase the generalizability of our study. Furthermore, these hospitals utilise the identical patient tracking software system. This information has been elaborated in the purpose section of our article.
- Statistical Analysis:The statistical methods are well-detailed, but clarifying why certain tests were chosen over others for specific data types could help readers understand the analytical decisions better.
A-4) Within the scope of your suggestions, we explained why we used those tests in statistical analysis.
- Results Presentation:
- Discussion of Anomalies or Outliers:If there were any unexpected results or outliers in the data, discussing these specifically could provide deeper insights into the ICU conditions and their impacts on patient prognosis.
A-5) Thank you for your suggestion. We would like to clarify that we made additions to the results section of our study based on the feedback from our referees and discussed them in more detail in the discussion section.
- Discussion and Conclusion Enhancements:
- Broader Implications:Expand the discussion on how these findings can influence ICU practices, policy-making, and future research. Providing specific recommendations based on the results could make the conclusions more impactful.
A-6) We included your valuable suggestions in the discussion section of our study. We have added information on how it can guide future ICU practices, policy, and research. Additionally, we have expanded our discussion section and provided more data about our results, supported by the literature.
- Limitations:While some limitations are mentioned, a more thorough examination of potential biases, the generalizability of the findings, and how these limitations could affect the interpretation of the results would strengthen the manuscript.
A-7) We would like to point out that the limitation section of our study has been completely revised in line with your valuable suggestions. We would like to point out that the weaknesses and strengths of our results have been added in terms of generalizability and interpretation.
- Future Research Directions:Suggest specific areas for future research that build directly on the findings of this study. This could include prospective studies, research in different geographic settings, or investigations into interventions to mitigate the adverse effects of high ICU workload on patient outcomes.
A-8) Thank you for your valuable suggestions and contributions. We would like to point out that we have made additions to both the discussion and conclusion parts of our study about future research and suggestions based on the results of our study.
- Technical and Language Corrections:
- Consistency in Terminology:Ensure consistency in using terms and definitions throughout the manuscript. For example, clarify if "tertiary ICUs," "training and research hospitals," and "City hospitals (CHs)" are used interchangeably or signify different categories.
A-9) We have taken your valuable suggestion on board and used more descriptive language throughout the article. Thank you very much.
- Grammar and Style:A thorough proofreading to correct grammatical errors and improve sentence structure would enhance readability. Ensuring the manuscript adheres to the journal's style guide is also crucial.
A-10) We would like to state that we received English proofreading support and certification for our article.
- References and Citations:
- Updating References:Check that all references are up-to-date and relevant. Including more recent studies, where available, could strengthen the argument and show the manuscript's current relevance.
- Citation Consistency:Verify that all citations are consistent in format and that all referenced works are included in the bibliography.
A-11) We have taken your valuable suggestions into account and made the necessary arrangements.
Implementing these suggestions could significantly enhance the manuscript's clarity, depth, and impact, making its contributions to the field more pronounced and actionable.
Yours sincerely

Reviewer 4 Report
Comments and Suggestions for Authors
The research is presented in a observational point of view, not as a research point of view. I think they need to look into some technical research aspects by which some research objectives may met.
Comments on the Quality of English LanguageThe research is presented in a observational point of view, not as a research point of view. I think they need to look into some technical research aspects by which some research objectives may met.
Author Response
Dear Editor,
We thank the reviewers for their valuable comments on the manuscript, and we have edited the manuscript to address their concerns. We have highlighted in red all the changed places in the main text. We would like to state that all the reviewers' comments have been taken into account and the necessary corrections have been made.
We hope that the article is now suitable for publication.
Yours sincerely
Semih AYDEMİR, Dr.
Yıldırım Beyazıt University Yenimahalle Teaching and Research Hospital, Department of Anaesthesiology and Resuscitation, Yenimahalle, Ankara, Turkey
On behalf of all authors.
Reply to Reviewer-4:
The research is presented in a observational point of view, not as a research point of view. I think they need to look into some technical research aspects by which some research objectives may met.
A) Thank you for considering our article. We have taken your suggestions into account and made revisions in every section based on the referees' recommendations. The article now presents a more research-oriented perspective. It should be noted that we conducted a new literature review for our study and made additions and revisions based on the suggestions. We hope that the corrections we made were appropriate. Our study examines the effects of intensive care conditions on mortality and morbidity in a multi-centred data set of 54,312 patients. It can make valuable contributions to the literature. Thank you for your interest and suggestions.

Round 2
Reviewer 4 Report
Comments and Suggestions for Authors
The paper substantially improves as it goes through the review process. English editing improves a lot. Most of the review remarks addresses successfully and written very well. I have no further objections. This paper could be accepted.
Author Response
Dear Editor,
We thank the reviewers for their valuable comments on the manuscript, and we have edited the manuscript to address their concerns.
We hope that the article is now suitable for publication.
Yours sincerely
Semih AYDEMİR, Dr.
Yıldırım Beyazıt University Yenimahalle Teaching and Research Hospital, Department of Anaesthesiology and Resuscitation, Yenimahalle, Ankara, Turkey
On behalf of all authors.
Reply to Reviewer-4:
The paper substantially improves as it goes through the review process. English editing improves a lot. Most of the review remarks addresses successfully and written very well. I have no further objections. This paper could be accepted.
- A) Thank you for your feedback and suggestions. Our article has received a lot of support from valuable reviewers. Thank you very much.
